# Insights into the recurrent energetic eruptions that drive Awu among the deadliest volcanoes on earth

Philipson Bani[1], Kristianto[2], Syegi Kunrat[2], Devy Kamil Syahbana[2]

1- Laboratoire Magmas et Volcans, Université Blaise Pascal - CNRS -IRD, OPGC, Aubière, France.

2- Center for Volcanology and Geological Hazard Mitigation (CVGHM), Jl. Diponegoro No. 57, Bandung, Indonesia

*Correspondence to*: Philipson Bani (philipson.bani@ird.fr)

**Abstract**

The little known Awu volcano (Sangihe island, Indonesia) is among the deadliest with a cumulative death toll of 11048. In less than 4 centuries, 18 eruptions were recorded, including two VEI-4 and three VEI-3 eruptions with worldwide impacts. The regional geodynamic setting is controlled by a divergent-double-subduction and an arc-arc collision. In that context, the slab stalls in the mantle, undergoes an increase of temperature and becomes prone to melting, a process that sustained the magmatic supply. Awu also has the particularity to host alternatively and simultaneously a lava dome and a crater lake throughout its activity. The lava dome passively erupted through the crater lake and induced strong water evaporation from the crater. A conduit plug associated with this dome emplacement subsequently channeled the gas emission to the crater wall. However, with the lava dome cooling, the high annual rainfall eventually reconstituted the crater lake and created a hazardous situation on Awu. Indeed with a new magma injection, rapid pressure buildup may pulverize the conduit plug and the lava dome, allowing lake water injection and subsequent explosive water-magma interaction. The past vigorous eruptions are likely induced by these phenomena, a possible scenario for the future events.

## 1 Introduction

Awu is a little known active volcano located on the Sangihe arc, northeast of Indonesia. It is the largest and the northernmost volcano of the arc with an aerial volume of ~27 km$^3$ that constitutes the northern portion of Sangihe Island (Fig.1). The edifice culminates at 1318 m above sea level and more than 3300 m from sea bed on its western flank (www.opendem.info). The summit crater is 1500 m in diameter and 380 m depth from from the highest point. The crater flow of 260 m in diameter is currently occupied by a cooling lava dome of 30 m height and 370 m in diameter, formed after the 2004 eruption. Since 1640, Awu went through 18 eruptions, including 2 of VEI 4 (Volcanic Explosivity Index; Newhall and Self, 1982), in 1812 and 1966. Such powerful VEI 4 events represent only 5% of the eruptions in the last 10,000 years (Pyle, 2015), and curiously two have occurred on Awu with a return period of 154 years. In the database of volcanic eruption victims compiled by Tanguy et al. (1998), Awu eruptions claimed a total of 5301 victims, including 963 casualties during the 1812 eruption, 2806 during the 1856 eruption and 1532 during the 1892 eruption.

This latter database did not take into account the 2508 victims of the 1711 eruption (Van Padan, 1983, Data Dasar Gunung Api 2011) and 3200 deaths following the 1822 eruption (Lagmay et al., 2007). The latest VEI 4 eruption on Awu in 1966 killed 39 people, injured 2000 and forced the evacuation of 420,000 inhabitants (Withan, 2005). In total since 1711, Awu's recurrent eruptive activities have caused a cumulative 11048 fatalities, mainly from lahar events. Awu is thus one of the deadliest volcanoes worldwide that merits better attention. In this works we aim to highlight the intense

eruptive character of Awu volcano and provide insights into the possible mechanisms i that fueled the deadly energetic eruptions.

**1.1 Geological setting**

The volcanism within the Molucca Sea is dominated by the unique example of the present-day arc-to-arc collision that involves the Sangihe arc from the west and the Halmahera arc from the east. The Molucca Sea plate that existed between the two arcs is currently dipping east under the Halmahera arc and west under the Sangihe arc. At least 600 km of lithosphere has been subducted to the west since its onset 20 Ma ago and on the opposite side, the Benioff zone associated with the east-dipping slab can be identified to a depth of 200-300 km (Hall et al., 1995; Hall and Wilson,

2000). This double subduction has led to the arc-arc collision that commenced around 3-5 Ma in the north of Molucca Sea and is currently considered as complete in the area north of Talaud island (Fig.1) (Cardwell et al., 1980). This is highlighted by the distribution of the 8 aerial active volcanoes only along the southern part of the 550 km of Sangihe arc. Beyond the Sangihe island, the volcanoes are inactive and dissected (Morrice et al., 1983), Awu being the northern most active volcano of the arc. Beneath the southern part Molucca Sea, the Sangihe forearc is presently overriding the

Halmahera forearc, while the Halmahera arc itself is thickening by the over-thrusting of its back-arc from the east (Hall and Wilson, 2000).

**1.2 Historical activities**

Since the early 1980s, numerous studies have pointed to Awu as the center of strong volcanic manifestations with global

impacts (Robock, 1981; 2000; Handler, 1984; Zielinski et al., 1994; Jones et al., 1995; Palmer et al., 2001; Donarummo et al., 2002; Guevra-Murua et al., 2015) although recent works have reviewed and declassified some of these events, including the 1641 event that was considered as responsible for 1642-1645 global cooling (e.g., Robock, 1981; Simkin et al., 1981; Jones et al., 1995) but later attributed to the Parker eruption of Jan. 4, 1641 (Delfin et al., 1997). Similarly, on the famous Edward Munch painting of 1893 - the "The Scream", the red sky was first considered as induced by the 1892

eruption of Awu (Robock, 2000) but was later attributed to Krakatau 1883 eruption (Olson et al., 2004) and then finally considered as inspired by the nacreous clouds (Fikke et al., 2017; Frata et al., 2018). But Awu's 1812 eruption of VEI 4 has loaded a significant amount of ash and aerosols into the atmosphere leading to the global abnormal correlation between dust load in the atmosphere and solar activity (Donarummo et al., 2002). In 1856, another eruption of Awu (VEI 3) injected massive amount of sulfate aerosols into the stratosphere, leading to an increase in the stratospheric aerosol's

optical depth, sufficient to reduce the sea surface temperature and thus subsequently reduce the number of tropical cyclones (Guevara-Murua et al., 2015). In contrast, Handler (1984) indicates that the 1966 eruption of Awu (VEI 4)

loaded a notable amount of aerosols into the stratosphere resulting in a warmer eastern tropical Pacific Ocean over three consecutive seasons with subsequent influence on El Nino type events. Such a regional response is typical of tropical major eruptions that produce asymmetric stratospheric heating (Robock, 2015). On the regional and local scale, Awu eruptive activities have triggered at least two tsunamis, on Mar. 2, 1856 and on Jun. 7, 1892 (Latter et al., 1981; Paris et al., 2014). No less than 18 eruptions were reported on Awu volcano since 1640, thus about 1 eruption every ~20 years, highlighted our compilation (Table 1). The latest eruption was a VEI 2 in 2004.

**2 Methodology**

The available documents that refer to Awu volcano, as summarized in the introduction, are generally incomplete but most point to vigorous explosions and subsequent casualties. Thus to gain more insights into Awu's volcanic activity, a reconnaissance visit to the summit was carried out in July 2015 with thermal and gas measurements. Thermal imaging was performed using OPTRIS PI400, a miniature infrared camera that weighs 320g, including a lens of 62°x49° FOV, f=8 mm and a dynamic range equivalent to the radiant temperature of -20°C to 900 °C. The detector has $382 \times 288$ pixels and the operating waveband is 7.5-13 µm. The maximum frame rate is 80 Hz. The camera was first positioned on the crater rim (Fig.2) observing the whole crater, then in the crater, looking at the two main hot surfaces (Fig.2). The radiant flux ($Q_{rad}$) estimation is obtained using the following: $Q_{rad} = A\varepsilon\sigma(T_s^4 - T_a^4)$, where A is the area of the hot surface, $\varepsilon$ is the emissivity (0.9 for andesite), $\sigma$ is the Stefan-Boltzmann constant ($5.67x10^{-8}$ W m$^{-2}$ K$^{-4}$), $T_s$ is the hot surface temperature and $T_a$ is the ambient temperature. Thermal results are corrected for an oblique viewing angle of 30° following the approach detailed in Harris (2013) and the hot surfaces in the crater are discriminated based on their brightness temperature ranges, including 16-20 °C, 21-25 °C, 26-30 °C, 31-35 °C and 36-41 °C. Such thermal ranges allowed better estimation of the total radiant flux given the heat distribution in the crater. Values below 16 °C fall in the background level whilst 41 °C is the maximum temperature observed from the rim. Temperature values were corrected for atmospheric influence relying on ACPC (Atmospheric Correction Parameter Calculator; https://atmcorr.gsfc.nasa.gov/) and validated with closer thermal recording before integrated into the radiant flux calculation. Thanks to the high acquisition rate of OPTRIS, a few series of continuous recordings were obtained on the most heated surfaces to retrieve the heat flow dynamics.

A portable Multi-GAS system from INGV (as used by Aiuppa et al. 2015; Bani et al., 2017; 2018) was deployed to measure the gas composition. The instrument was positioned in the main degassing point at the northern part of the crater (Fig.2) and simultaneously acquired concentrations of $H_2O$, $CO_2$, $SO_2$, $H_2S$, and $H_2$ at 0.1 Hz. Data were processed using Ratiocalc (Tamburello 2015). The scanning DOAS was used for the gas emission budget. The instrument performed at fixed position in the crater (Fig.2). Further details on Awu gas measurements are provided in a separate paper and hereafter referred to as (Bani et al. submitted).

During this fieldwork, a less altered rock sample was selected directly on the lava dome then analyzed for major and trace elements using ICP-AES. The same sample was analyzed in two different laboratories, including Laboratoire Magmas et Volcans (Clermont-Ferrand, France) and Pôle de Spectrométrie Océan (Brest, France).

**3 Results**

The whole-rock composition of the lava dome, obtained from ICP AES analysis, indicates a dome composition of 52-56% $SiO_2$ and relatively low alkali contents corresponding to a basaltic-andesite (Table 2, Fig.3). Results are comparable with the data from Morrice et al (1983) and Hanyu et al. (2012). Trace elements normalized to N-MORB point to elevated ratios of large ion lithophile elements (LILEs), light rare earth elements (LREEs) and high-field strength elements (HFSEs) (Table 2, Fig.3).

DOAS measurement results obtained on Awu indicate a relatively small degassing with a mean daily $SO_2$ emission rate of 13±6 tons. The multigas results indicate $H_2S/SO_2$, $CO_2/SO_2$, $H_2/SO_2$ and $H_2O/SO_2$ ratios of 49, 297, 0.1 and 1596 respectively with the corresponding gas composition equivalent to 82% of $H_2O$, 15% of $CO_2$, 2% of $H_2S$, 0.05% of $SO_2$ and 0.02% of $H_2$. Assuming the above results are representative then $H_2O$, $CO_2$, $H_2S$, and $H_2$ emission rates would be 5800 t/d, 2600 t/d, 340 t/d and 0.1 t/d respectively. The gas equilibrium temperature obtained by resolving together the $SO_2/H_2S$ vs. $H_2/H_2O$ redox equilibria (see methodology in Aiuppa et al., 2011; Moussallam et al., 2017) is circa 380 °C (Bani et al., submitted)

Thermal infrared recording from the rim highlights two main heated surfaces in Awu's crater, but both are located in the northern part of the lower crater wall, next to the lava dome (IR2, IR3, Fig.4). It is also evident from these thermal results that the lower crater wall around the dome is much hotter than the lava dome itself. The total radiant flux from the crater (IR1, Table 4) is 27±12 MW, including 5.6±2.4 MW from the lava dome and 21±9 MW from the area surrounding the dome. The highest radiant flux per area (0.9 MW) is recorded in the IR3 zone where gas is released at a low frequency of 0.3 Hz with a thermal fluctuation amplitude of 0.1-0.3 MW.

**4 Discussion**

**4.1 Melt source**

To sustain the recurring strong eruptive activity of Awu, highlighted by one strong eruption every 20 years over the last 3.5 centuries (including two VEI 4 and three VEI 3), requires sufficient magma supply rate. The total alkaline vs. $SiO_2$ diagram (Table 2., Fig.3) indicates a basaltic andesite magma, typical of island arc volcanoes where the geodynamic context allows a relatively evolved magmatic source. Awu is part of the Sangihe arc where the geodynamic processes are controlled by the divergent double subduction that resulted in the Sangihe forearc overriding the Halmahera forearc (Cardwell et al.,1980, Morrice et al., 1983, Hall and Wilson, 2000; Jaffe et al., 2004; Zhang et al., 2017; Bani et al., 2018). The pattern obtained by normalizing the trace elements to N-MORB indicates high LILE (Cs, K, Rb, Ba and Sr) content and low abundance of HFSE, represented by Nb, typical of subduction melt source in which the mantle wedge has been contaminated by fluid released from the subduction slab (McCulloch and Gambke, 1991; Davidson 1996, Mcpherson et al., 2003). This result is coherent with Jaffe et al. (2004) who highlight low $^3He/^4He$ (5.4-6.4 $R_A$) and high $CO_2/^3He$ ratios (64-180 x$10^9$) as well as high $\delta^{13}C$ (≥-2‰) suggesting slab contribution into the magmatic fluids at Awu. Clor et al. (2005) further point out anomalous high $N_2/He$ (2852) coupled with low $\delta^{15}N$ (3.3%) suggesting increased slab contribution, possibly by slab melting as collision stalls the progress of the subducting plate and allows it to become superheated (Peacock et al., 1994). This is supported by the slow-down of the subduction rate as evidenced by seismic studies (McCaffrey, 1983; Pubellier et al., 1991; Zhang et al., 2017). This particular double subduction and arc-arc

collision have rendered the slab prone to melting (Clor et al., 2005) that subsequently supply the magmatic source beneath Awu volcano.

## 4.2 A conduit plug

Lava domes are formed when viscous lava extrudes to the surface effusively then piles up around the vent. Such
phenomena involve complex processes, including crystallization, bubble nucleation, growth, coalescence and out-gassing, bulk magma deformation, crack propagation and healing (e.g., Ashell et al., 2015 and ref therein). It is the competition between these processes that either promotes or prevents degassing, leading to explosions or stability of a lava dome (Klug and Cashman, 1996; Takeuchi et al., 2005; Mueller et al., 2008). On Awu, the $SiO_2$ content of the lava dome higher than 50 wt% as well as the perfect semi-spherical morphology of ~$1.3x10^7$ m$^3$ that extended from the
middle of the crater suggests an endogenous growth that generally inflates the dome carapace through magma injection at depth. In such case, lava domes are known to induce variable porous and brecciated carapace surrounding a denser and coherent interior (Newhall and Melson, 1983; Fink et al., 1992; Wadged et al., 2009, Ashell et al., 2015) suitable to form a plug in the upper conduit (Watts et al. 2002). The radiant thermal energy around the lava dome is much higher than the heat release from the dome representing 79% of the total 27 MW from the crater. Only ~6 MW is released
through the lava dome itself. It is also around the dome that much of the gas is released to the atmosphere (Figure 5). The hottest surfaces also correspond to the main degassing points which suggests that heat is rather sustained by fluid circulations around the dome. With a conduit plug, the gas released at depth is thus forced to the periphery of the lava dome (Fig.5), similar to other dome-forming systems, including Rokatenda (Primulyana et al., 2018), Lascar (Matthews et al., 1997) or Soufriere Hills (Sparks, 2003).
It is thus obvious that the existence of a conduit plug may constitute a barrier to the gas flow, suitable for rapid pressure build up with new magma injection, a situation that can strongly contribute to the vigorous explosions on Awu.

## 4.3 The heat transfer to the surface controls the water accumulation

Out of the 18 recorded eruptive activities on Awu, 11 were tagged as phreatic and 7 other eruptions were
170 phreatomagmatic and magmatic (Table 1). It is thus unambiguous that water played a major role in Awu volcanic activity. Indeed, with an average annual rainfall of 3500 mm (Stone, 2010) and a crater area of 1.5 km$^2$, the Awu summit is likely to accommodate $5.2x10^6$ m$^3$ of water each year. Given that there is no visible water outlet from the crater, one can expect water accumulation and strong infiltration into the hydrothermal system which may then subsequently contribute to phreatic eruptions. But as highlighted in figure 6, surface water was not always present in Awu's crater. A
175 crater lake existed in 1922, 1973 and 1995 whilst in 1931 and 1979 a crater lake co-existed with a lava dome. In July 2015 (this fieldwork) there was no water in the crater and a lava dome occupied the central part of the crater. July is among the driest months of the year, however, the average monthly rainfall on Sangihe Island doesn't fall below 130 mm (Stone, 2010). Hence one can expect a cumulative water volume of at least $195x10^3$ m$^3$ (equivalent to $7x10^6$ moles or $1.3x10^8$ g, using PV=nRT) into Awu's crater during that period of the year. But the absence of water as observed in July
indicates that the water was efficiently infiltrated and evaporated away. In theory, if we assume that the infiltration is negligible, then it requires a heat energy of $8.0x10^{11}$ joules (using $mC_p\Delta T$; m is the water mass, Cp is the water's specific

heat capacity, ΔT is the difference between boiling and ambient temperature (taken as 16°C from IR camera since no meteorological data is available) to bring the above volume to the evaporation temperature (100°C) and another $4.6 \times 10^9$ joules  (using mL; L is the latent heat of vaporization) to convert it into water vapor. A total $8.1 \times 10^{11}$ joules is thus sufficient to dry out the July incoming water volume. With 27 MW of radiant flux from the crater, equivalent to $2.7 \times 10^7$ J s$^{-1}$, only 8 hours is necessary to heat the $7 \times 10^6$ moles of water from 16 °C to the evaporation temperature and transform it to water vapor. This duration should be considered maximum as the portion of water infiltration is ignored. Nevertheless, the above simple calculation suggests that the heat transfer to the surface from the magmatic source is largely sufficient to evaporate out the water and thus the water accumulation in Awu's crater could rely on the amount of heat supply by the shallow magma body.  In 1992, Awu's crater lake experienced 95% of water loss from its initial volume  of $3.5 \times 10^6$ m$^3$  (Table 1). Although attributed with no details to a seepage through active faults beneath the crater (GVP, 2004), the decrease of water pH from 5 to 3 and the lake edge temperature of ~40°C (GVP, 1992) indicate a possible fluid supply from depth that subsequently led to the October 1992 phreatic eruption (Table 1). During such event water loss through evaporation can be non negligible. This was the case during the 1995 phreatic eruption in lake Voui where more than 14 million cubic meter of water was lost through evaporation (Bani et al., 2009), whilst the lake edge temperature was 40°C and a pH of 2-3 (Wiart, 1995). It thus likely that the heat supply by the combined hydrothermal and magmatic system has contributed to the significant water loss on Awu in 1992.  Similarly, in 2004 the lake water progressively dried out before the eruption, in response to the progressive increase of heat flux that ultimately reach 38.8 MW on Jun. 8, 2004 (http://modis.higp.hawaii.edu/), much higher than the 27 MW obtained in Jul. 2015 (Table 3).   If the increase of heat flux can lead to water lake evaporation, the cooling of the crater surface can in contrast allow water to accumulate. Hence assuming that the current cooling trend in Awu's crater continues, then ultimately the heat supply to the surface will no longer be sufficient to dry out the incoming water from the high annual rainfall. Water may then accumulate to form a new crater lake, as already witnessed in the past.

**4.4 The Hazardous situation**

The contact of liquid water with a hot surface is widely accepted as a process that can trigger explosive water-magma interactions (Wohletz, 1986; 2002; Zimanowski et al., 1995; Thiéry and Mercury, 2009). However, according to a review of historical eruptions through volcanic lakes, ~2% involved relatively passive growth of subaqueous to emergent lava domes (Manville, 2015). This was the case at Kelud volcano (Java) in 2007 when a lava dome emerged in the middle of a crater lake creating a threat of a major eruption. But the vigorous explosion never happened as suspected (Hidiyati et al., 2009). Only 7 years later, in 2014 a VEI 4 eruption was witnessed at the same volcano (Kristiansen et al., 2014; Caudron et al., 2015). Such a delay between dome occurrence through crater lake and major eruption was also witnessed on Awu in 1996 where a VEI 4 event occurred 35 years after the lava dome emplacement (in 1931). The same scenario was repeated in 1992, when an eruption occurred 13 years after another lava dome passively erupted through the crater lake. It is thus likely that during ascent, the crystallizing magma has released much of its gas and the carapace surface temperature has rapidly cooled below 100°C once reached the surface (Sherrod et al., 2008). In such scenario the dome may passively emerged through a crater lake without explosive magma-water interactions.  In 2004, the lava dome formation on Awu differs from the past dome emplacements. There was no crate lake and its growth constituted the last

event of the 2004 eruptive event. In less than two weeks the dome reached its current size then it completely stopped from growing. Hence, whatever the scenario is, lava dome emplacement on Awu did not immediately trigger eruption.

Furthermore, the current cooling lava dome  on Awu was developed on a flat crater floor which is more than 350 m below the crater rim. The chances to witness a dome collapse on Awu are thus negligible in contrast to other lava dome emplacements, were the average growth rate is approximately $10^4$ m$^3$ day$^{-1}$ with a mean volume of $5x10^7$ m3, some with unstable slopes (Newhall and Melson, 1983). The hazardous situation on Awu is rather related to the presence of the conduit plug and a crater lake since this latter may further increase the potential of violent eruptions (Sheridan and Wohletz 1983; Wohletz 1986). Based on the size of Awu's lava dome, the subsurface volcanic system has to develop more than 3.1 MPa of pressure (Pressure (Pa) = F/Area (m$^2$); F (N) = masse (kg) x 9.8 (m/s$^2$); density of 2700 kg/m$^3$) to destabilize the $3.5x10^{10}$ kg of lava dome. However, given that tens of mega-pascals can be easily developed in the conduit or reservoir (e.g., Gudmundsson, 2012),  there is no doubt that the system will clear-up the conduit and pulverize the lava dome in the future, as already witnessed in the past. The common process that drives eruptive activity is the injection of a new magma into a subvolcanic reservoir (Pallister et al., 1991; Williamson et al., 2010). Indeed, besides adding volume that increases the overpressure of the magma on the confining walls, the heat introduced by the process induces convection and vesiculation (Sparks et al., 1977) that subsequently mobilize the crystal-rich magmas (Burgisser and Bergantz, 2011), whilst the fluxing of volatiles increases the buoyancy (Costa et al., 2013; Williamson et al., 2010; Parmigiani et al., 2016). All these mechanisms concur to pressure buildup and eventual eruption.  Alternatively, given the degassed magma source on Awu, highlighted by the prevalence of $H_2S$ over $SO_2$, the low $SO_2$ emission budget (13 t/d) and the low equilibrium temperature of circa 380 °C obtained by resolving together the $SO_2/H_2S$ vs. $H_2/H_2O$ redox equilibria (Aiuppa et al., 2011; Moussallam et al., 2017), a second crystal nucleation event at shallow depth can occur (Melnik and Sparks, 2005). The rapid crystallization that follows will lead to an over-saturation of the remaining melt with intense diffusion of volatiles and growth of bubbles (Cashman, 1992; 1998; Swanson et al., 1989). The process can lead to powerful eruptive discharge that can possibly clear out the conduit plug and the lava dome. This was the mechanism that pulverized the lava dome at Kelud in 2014 (Cassidy et al.2018) and pumped out the degassed magma on Rokatenda in 2012-2013 (Primulyana et al., 2017). Another mechanism that can possibly induced eruption on Awu, given the predominant hydrothermal manifestation, is the acidic-sulphate alteration (Heap et al., 2019). This latter reduces the permeability of the lava dome which in turn promotes pore pressure increases that can eventually lead to eruption. Whatever the triggering mechanism,  the future eruption will clear out the conduit plug and pulverize the lava dome, as already witnessed in the past. If such event occur in the presence of a crater lake,  the large influx of water into the conduit could lead to explosive magma-water interaction. The intensity of such mechanism depends on the efficacy of the heat transfer from the magma to water which is directly correlated to the size of the contact area between magma and water. This latter is likely to increase significantly and the heat transfer rate may escalate with the conduit plug clearing. Such a great energy release may induce explosive vapor expansion, thorough magma fragmentation and subsequently the formation of convecting columns, a significant and more widespread tephra dispersal through fall and possibly pyroclastic density currents. Such events of particularly high intensity are described as phreato-Plinian eruptions and can only be explained with the involvement of surface water in the eruption dynamics (Areva et al., 2018).

The presence of sufficient water volume in the crater besides the lava dome may thus constitutes the most hazardous situation at Awu volcano (Fig.7), a possible scenario behind the past vigorous eruptions.

**5 Conclusion**

Awu is the northernmost active volcano of the Sangihe arc. It is the center of 18 eruptive activities over the last 3.5 centuries, some with very strong intensity (VEI 4 and VEI 3). The pyroclastic and lahar events triggered by these eruptions have killed a cumulative 11048 inhabitants of Sangihe island, highlighting this volcano as one of the deadliest on Earth. Paradoxically, very little is known about this volcano. As emphasized in this work, the regular magma supply that sustained the activity of Awu is possibly linked to the peculiar geodynamic context of the region, controlled by the

divergent double subduction and the subsequent arc-arc collision. Awu also has the particularity to host alternatively or simultaneously a lava dome and a crater lake. Lava domes seem to erupt passively through a crater lake and the heat that accompanied this emplacement has shown to be sufficient to dry out the lake. The emplacement of the lava domes also appears to be associated with a conduit plug development that forces the degassing to the crater wall. With time the lava dome cools down, allowing a progressive crater lake formation with the high annual rainfall in the region. This scenario

may ultimately constitute the most hazardous situation if a new magma injection occurs at depth, or the system undergoes a second crystal nucleation event within the degassed magma or the lost of permeability of the lava dome with the acidic-sulphate alteration. Indeed these latter processes may induce a rapid pressure buildup that can pulverize the conduit plug and the lava dome, creating a favorable condition for a significant water injection and a subsequent explosive water-magma interaction. Such a scenario likely resulted in the past vigorous eruptions on Awu and may occur

in the future given the presence of a cooling lava dome and a conduit plug.

**Acknowledgments**

This work was supported by IRD under the JEAI-COMMISSION program in collaboration with CVGHM. Sincere acknowledgment to pak Endi and the other staffs of Awu observatory for their support in organizing the expedition to the

crater of Awu. We thank Dr. Corentin Caudron  and Dr. Caroline Bouvet de Maisonneuve for their detail review and helpful comments that substantially improve this manuscript.

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

**Captions**

**Table 1.** History of Awu eruptive activity. Most of the information is obtained from Data Dasar, Gunung Api, (2011) and Siebert et al. (2010).

| Date | Eruptive events |
|---|---|
| 1640 (Dec.) | Phreatic eruption (Data Dasar, Gunung Api, 2011; Siebert et al., 2010). |
| 1641 (Jan.3-4) | Phreatic eruption, lahar event (Wichmann, A., 1893; Siebert et al., 2010; Data Dasar, Gunung Api, 2011). |
| 1677 | Phreatic eruption (Data Dasar, Gunung Api, 2011). |
| 1711 (Dec. 10-16) | On the night of Dec. 10, violent eruption (VEI 3) propelled incandescent material above the summit. Pyroclastic flow combined with hot lahar, generated by the outburst of the crater lake, wiped out the entire city of Kandhar located at the eastern base of the edifice. About **3000** people were killed, including 2030 in Kendhar and 408 at Tahuna. Among those victims, 400 corpses were described as suffocated by the heat of pyroclastic (Wichmann, A., 1893; Van Padang, 1983; Data Dasar, Gunung Api, 2011; Siebert et al., 2010). |
| 1812 (Aug. 6-8) | Large phreatomagmatic eruption (VEI 4) with manifestations comparable to the 1711 event. Lahar and pyroclastic flows have destroyed villages, destroying all the coconut trees along the coast. **963** inhabitants were killed, particularly in the village of Tabuhan, Khendar and Kolengan (Tanguy et al., 1998; Data Dasar, Gunung Api, 2011). |
| 1856 (Mar. 2-7) | Large phreatomagmatic eruption (VEI 3) with associated pyroclastic and lahar flow that killed 2806 inhabitants. The eruption has also triggered a tsunami event (Wichmann, A., 1893; Siebert et al., 2010, Tanguy et al., 1998). |
| 1875 (Aug.) | Phreatic eruption (VEI 2) with no further report (Siebert et al., 2010; Data Dasar, Gunung Api, 2011). |
| 1883 (Aug. 25-26) | Eruption (VEI 2) but no further detail (Siebert et al., 2010). |
| 1885 (Aug. 18) | Phreatic eruption (VEI 2) but no further detail (Siebert et al., 2010;  Data Dasar, Gunung Api, 2011). |
| 1892 (Jun. 7-12) | Large phreatomagmatic eruption (VEI 3). Beginning at 6:10 am – then a huge column was seen ascending into the atmosphere in the afternoon, accompanied by lightning and thunderstorms. Muddy rain turned into pumice and heavy ashfall when the eruption reached the climax of its violence at 9 pm with pyroclastic flow and lahar before it started to fade after midnight. A large number of huts collapsed under the weight of ash and extensive mudflow occurred during and following the event. The eruption has also triggered a tsunami event. 1532 inhabitants were |

| | |
|---|---|
| | reported killed, mainly by pyroclastic and lahar events in many areas, including Mala, Akembuala, Anggis, Mitung, Kolengan, Metih, Khendar and Trijang. Many victims are killed while in church buildings (Wichmann, A., 1893; Van Padang, 1983; Data Dasar, Gunung Api, 2011; Tanguy et al., 1998). |
| 1893 | Phreatic eruption (VEI 2) but no further detail (Data Dasar, Gunung Api, 2011; Siebert et al., 2010). |
| 1913 (Mar. 14) | Phreatic eruption (VEI 2) (Data Dasar, Gunung Api, 2011; Siebert et al., 2010). |
| 1921 (Feb.) | Pheatic eruption (VEI 0) – crater lake activity (Data Dasar, Gunung Api, 2011; Siebert et al., 2010). |
| 1922 (Jun.-Sep.) | Pheatic eruption (VEI 0) - crater lake activity (Data Dasar, Gunung Api, 2011; Siebert et al., 2010). |
| 1931 (Apr.-Dec.) | Lava dome started to form in the crater lake in April and then progressively grew until reaching 80 m above the water in Dec. (Data Dasar, Gunung Api, 2011). |
| 1966 (Aug. 12) | At 8:20 (Aug.12), a VEI 4 began with a sudden thick smoke that rose from the crater associated with a strong blast. An hour later another strong blast occurred propelling voluminous amount of ash that subsequently blanketed the summit. Other strong explosions followed until around 13:30 and pyroclastic flow extended 5 km from the crater. Lahars have traveled 7 km toward the coast. along the water channels. Both phenomena have destroyed everything in their respective passages. Kendhar and Mala were the most affected area with 13 and 18 casualties respectively. Eight other inhabitants were also killed in other areas, including 2 officials. In total, the eruption killed 39 and caused the displacement of 11000 inhabitants (Data Dasar, Gunung Api, 2011; Siebert et al., 2010). |
| 1992 (May-Oct. 12) | Phreatic eruption (VEI 1). Before the eruption, the lake volume decreased by 95% from the initial $3.5 \times 10^6$ m$^3$ of water. On Oct. 12, a phreatic eruption occurred (Data Dasar, Gunung Api, 2011; Siebert et al., 2010). |
| 2004 (Jun. 8-10) | Magmatic eruption (VEI 2) building a column of 1000-3000 meters above the crater. The resulting ashfall extended kilometers from the volcano. At Tabukan, 15 km southeast of the volcano the ash was 0.5-1 mm thick. 18648 people were displaced but no one was killed (Data Dasar, Gunung Api, 2011; Siebert et al., 2010). |

**Table 2.** Major and trace composition of Awu lava dome.

| | S1 (lava dome) | S2 (lava dome) | S3* (volc. rock) | S4* (south Sangihe Is.) | S5** | S5** |
|---|---|---|---|---|---|---|
| $SiO_2$ (wt%) | 53.95 | 54.10 | 52.27 | 58.50 | 55.11 | 54.54 |
| $TiO_2$ | 0.77 | 0.71 | 0.72 | 0.61 | 0.69 | 0.73 |
| $Al_2O_3$ | 18.40 | 18.90 | 18.44 | 18.26 | 18.73 | 18.68 |
| $Fe_2O_3$ | 8.85 | 8.60 | 8.20 | 7.47 | 8.53 | 8.52 |
| $MnO$ | 0.21 | 0.20 | 0.19 | 0.16 | 0.19 | 0.19 |
| $MgO$ | 3.79 | 3.61 | 3.44 | 2.61 | 3.34 | 3.42 |
| $CaO$ | 8.73 | 8.85 | 8.42 | 7.16 | 7.76 | 8.60 |
| $Na_2O$ | 3.70 | 3.53 | 3.91 | 3.51 | 3.52 | 3.51 |
| $K_2O$ | 1.17 | 1.13 | 1.21 | 1.50 | 1.40 | 1.21 |
| $P_2O_5$ | 0.19 | 0.16 | 0.19 | 0.22 | 0.22 | 0.19 |
| LOI | 0.30 | 0.0 | 0.34 | 0.94 | | |
| Total | 99.75 | 99.83 | 98.69 | 100.65 | 99.49 | 99.59 |
| Ba (ppm) | 227 | 230 | 228 | 214 | 272 | 226 |
| Sr | 378 | 378 | 373 | 431 | 419 | 366 |
| Rb | | 19 | 20 | 30 | 25.5 | 19.80 |
| Cs | | | 1.0 | 0.7 | 1.00 | 1.01 |
| Y | | 23.3 | 26 | 51 | 18.8 | 20.5 |
| Zr | | 70.7 | 74 | 107 | 72.8 | 72.0 |
| Ni | | 5.4 | <5 | <5 | | |
| Sc | | 22.4 | | | | |
| V | | 242.5 | | | | |
| Cr | | 4.9 | | | | |
| Co | | 19.4 | | | | |
| Nb | | 2.6 | | | 3.04 | 2.17 |
| La | | 4.9 | | | 7.25 | 5.95 |
| Ce | | 13.8 | | | 16.4 | 14.80 |
| Nd | | 8.7 | | | 10.8 | 10.30 |
| Sm | | 2.3 | | | 2.90 | 2.95 |
| Eu | | 0.72 | | | 0.97 | 1.01 |
| Gd | | 2.7 | | | 3.20 | 3.45 |
| Dy | | 2.8 | | | 3.43 | 3.80 |
| Er | | 1.7 | | | 2.20 | 2.5 |
| Yb | | 1.92 | | | 2.20 | 2.48 |
| Th | | 0.4 | | | 1.33 | 1.10 |

* data from Morrice et al. (1983); ** data from Hanyu et al. (2012).

**Table 3.** Thermal radiant flux from Awu crater

| | | IR1 | IR2 | IR3 | Dome |
|---|---|---|---|---|---|
| **Surface (m2)** | | 92681 | 1852 | 1631 | 13225 |
| **Temp. range** | **Corrected mean temp. (°)** | **Surface occupied per temperature range (%)** | | | |
| **16-20 °C*** | 93.4 | 51.8 | 54.2 | 51.3 | 78 |
| **21-25 °C** | 97.3 | 1.1 | 27.9 | 13.9 | 0.5 |
| **26-30 °C** | 101.2 | 0.4 | 11.4 | 13.5 | 0.1 |
| **31-35 °C** | 105.3 | 0.3 | 6.5 | 14.3 | 0 |
| **36-41 °C** | 109.8 | 0.1 | 0 | 2.9 | 0 |
| **Mean Radiant flux (MW)** | | **27 ± 12** | **0.5 ± 0.2** | **0.9 ± 0.3** | **5.6 ± 2.4** |

*\* Note that below 16°C, it was difficult to discriminate the heated zones from the background surface.*

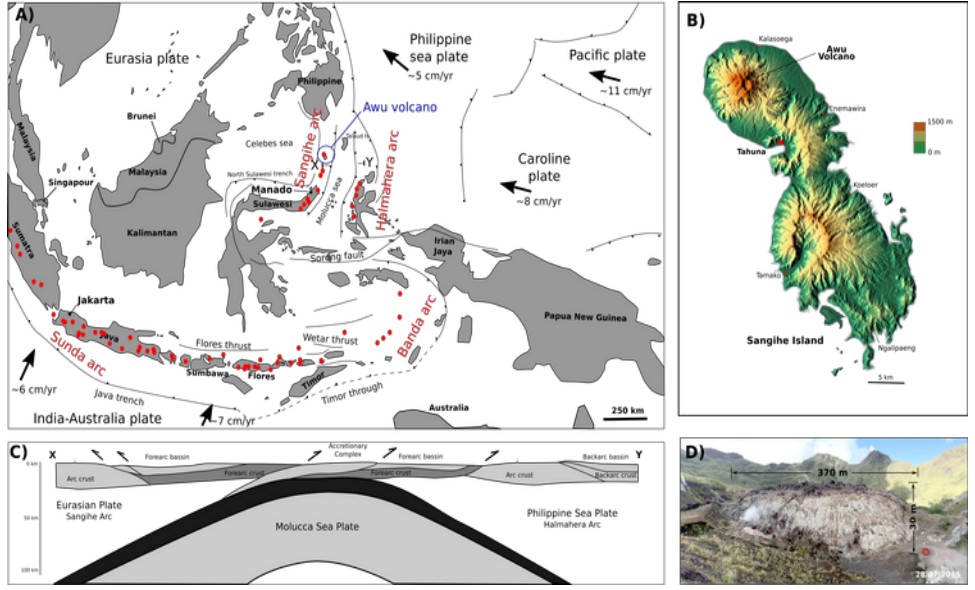

**Figure 1.** Awu volcano is the northernmost active volcano of the Sangihe arc (A). It occupies the northern portion of Sangihe island (B). 3D map from https://maps-for-free.com. Sangihe and Halmahera arcs constitute the present-day example of arc-to-arc collision (C). The Molucca Sea Plate that existed between the two arc is now sinking deeper beneath the Molucca Sea. Awu's crater is currently occupied by a lava dome (D). Note the person circled in red for scale.

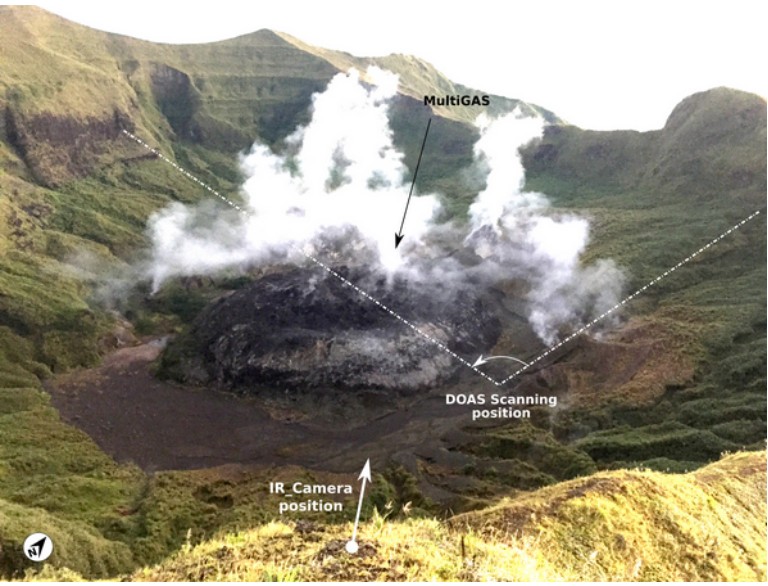

**Figure 2.** Awa lava dome in the crater. Degassing occurs from the lower crater wall and the northern part of the crater is the main degassing zone. The positions of DOAS scanning, MultiGAS (MG) and Infrared Camera are highlighted. The arrow of the IR_cam denotes the direction of the thermal camera.

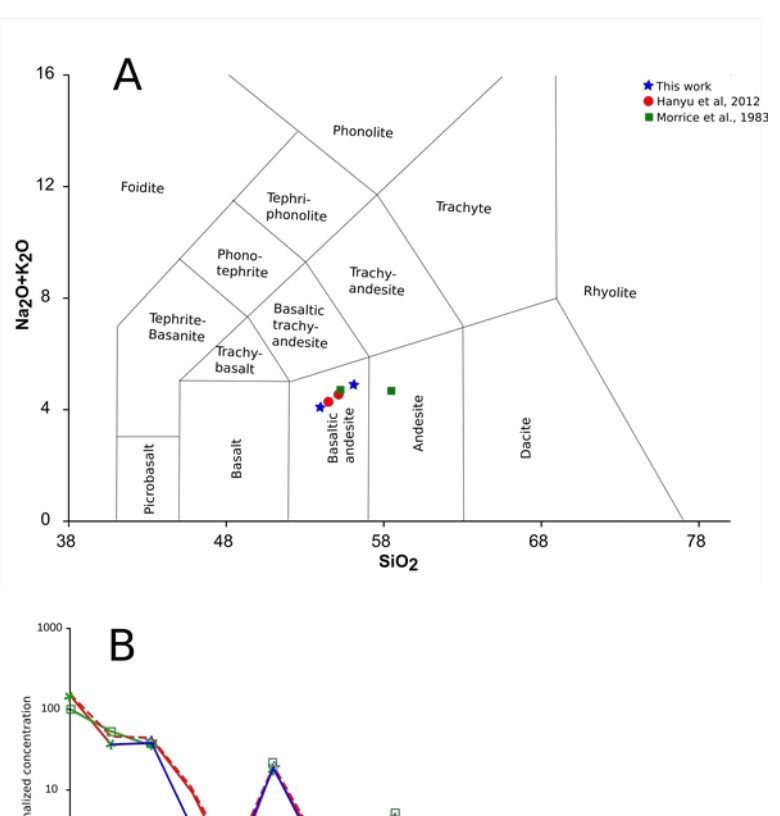

**Figure 3.** (A) Awu melt source is of basaltic andesite composition. Note that the sample from the southern part of Sangihe island (Morrice et al., 1983) rather indicates an andesite source. (B) Trace elements normalized to N-MORB indicate elevated ratios of LILE, LREE HFSE.

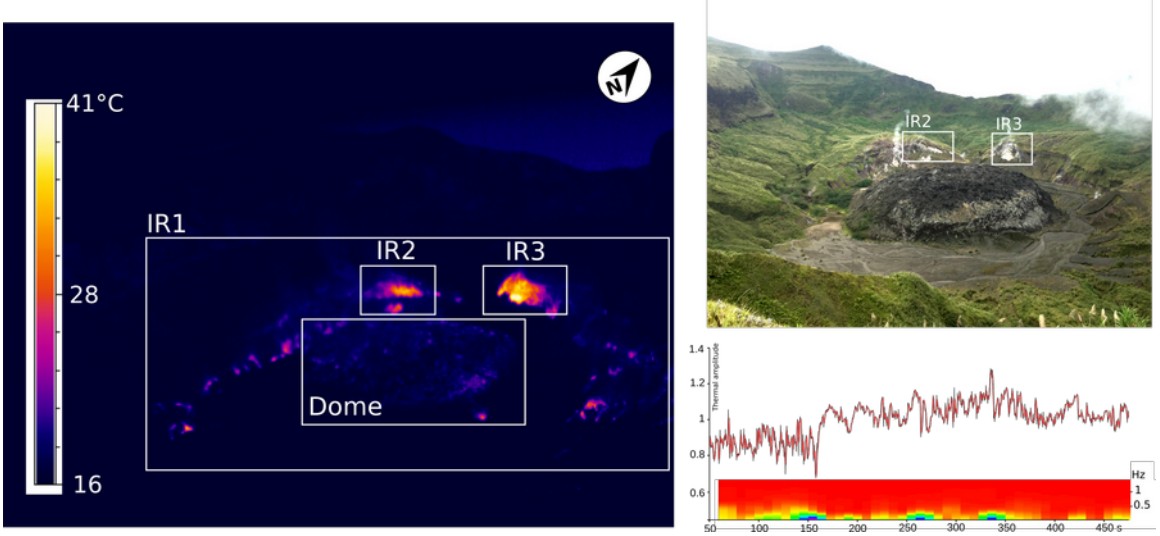

**Figure 4.** Thermal image highlighting the two most heated surfaces in the crater, as well as the lava dome being less hotter than the surrounding surface. White rectangles (IR1, IR2, IR3 and Dome) are the zones of interest in the radiant flux calculation (Table 3). The picture on the right gives a global view of the dome and its surroundings. The continuous thermal recording highlights a degassing dynamic through the IR3 zone characterized by a thermal fluctuation amplitude of 0.1-0.3 MW and at 0.3-0.4 Hz.

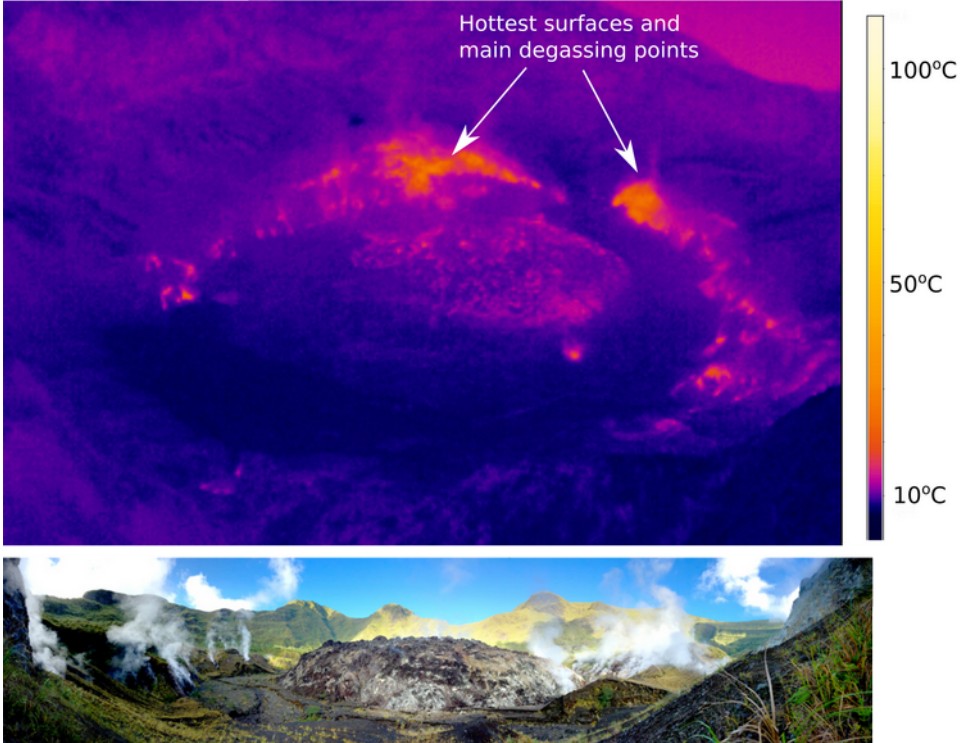

**Figure 5.** The lava dome and the conduit plug force the degassing to the crater wall. The hot surfaces (photo above) also correspond to the degassing points (picture below).

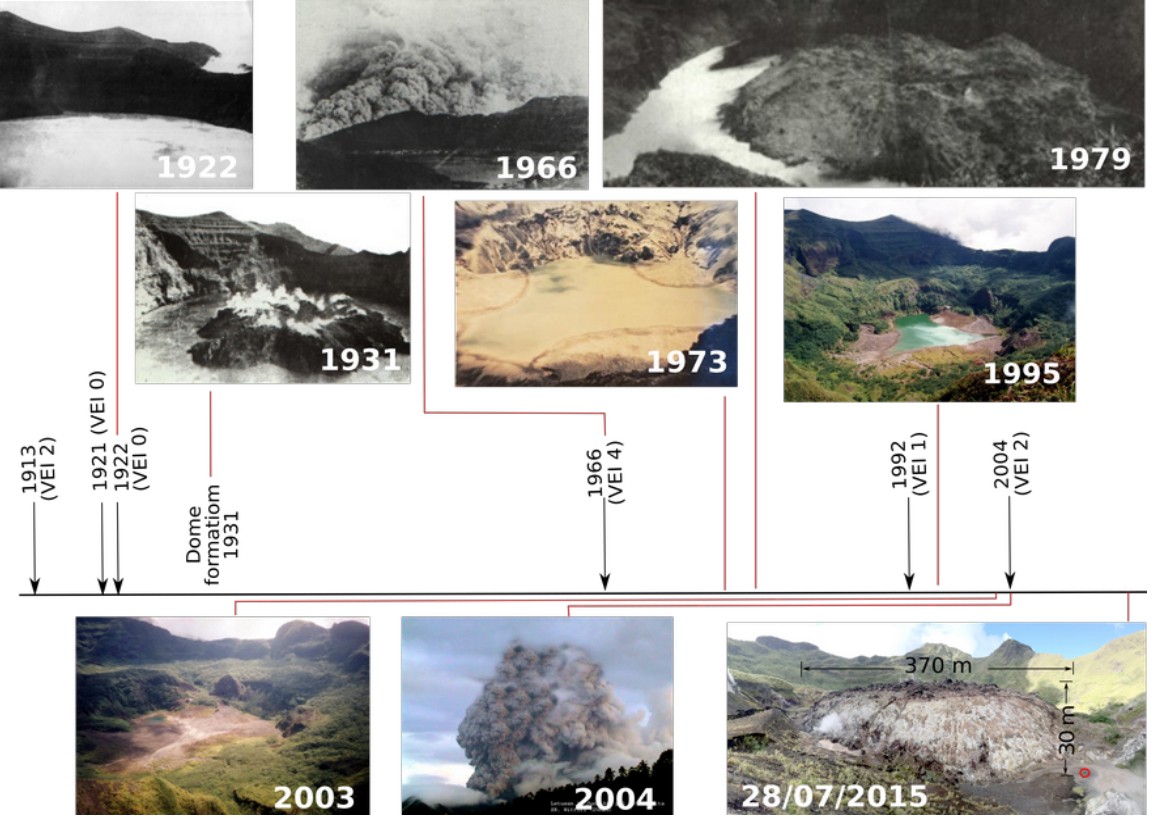

**Figure 6.** The configuration in Awu's crater is subjected to evolve from crater lake to lava dome emplacement and also from unoccupied crater to coexistence of lava dome and crater lake. Pictures from GVP, 2013 (Awu) except the 2015 (this work).

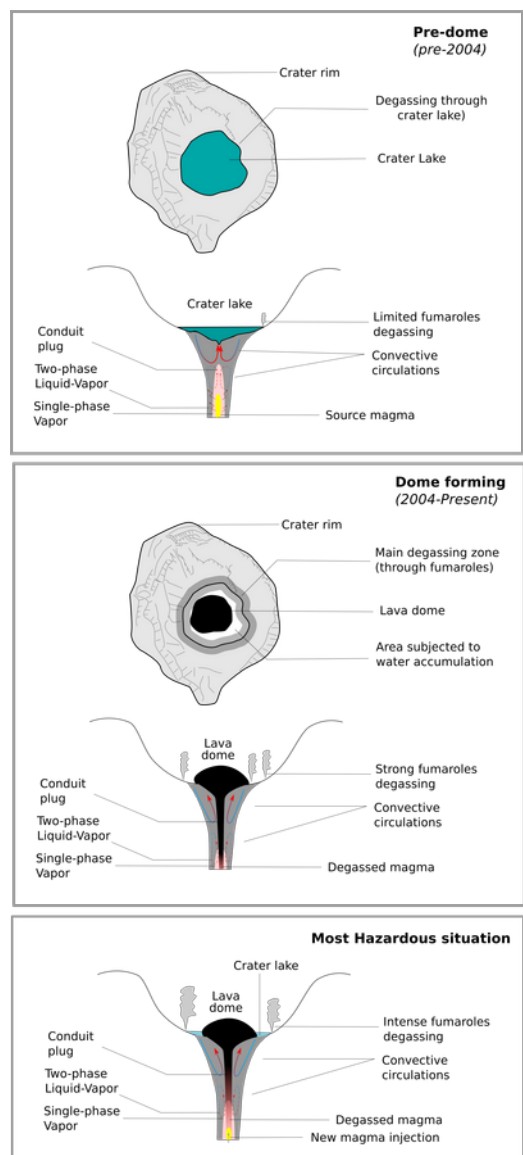

**Figure 7.** Pre-dome situation: A crater lake in Awu's crater. Few fumaroles on the crater wall. Dome forming situation: Lava dome emerged at the surface leading to lake water dry-out. Numerous fumaroles around the lava dome due to conduit plug. Most Hazardous situation: The cooled lava dome allowed the formation of a new crater lake. The new magma injection may enhance the opening of the conduit plug, leading to subsequent explosive water-magma interaction.