# Peer review of "Insights into the recurrent energetic eruptions that drive Awu among the deadliest volcanoes on earth"

_Natural Hazards and Earth System Sciences, 2020_

## Referee Comment (RC1) · Corentin Caudron (Referee) · 26 Mar 2020

**Corentin Caudron (Referee)**

corentin.caudron@univ-smb.fr

Received and published: 26 March 2020

This paper provides insights into the Awu volcano; a poorly known but dangerous volcano located in Indonesia. The authors collected and analysed thermal IR, Multi-GAS and petrological data to identify the main hazards associated with this volcano. They also compiled information on past eruptions.

The manuscript is overall well-written and presents interesting data and photos on this interesting system. I'm finding this paper suitable for publication but still have general comments and questions, and identified several technical problems (typo, etc.) and mistakes. Some important statements do not seem supported at this stage by the
results, but it might just be a matter of rephrasing. In any case, some clarification and further discussion is required (see below). I hope my review will help improve the manuscript and foster discussion.

Specific comments The introduction just documents past eruptions. It would be interesting to briefly introduce your aim and which methods you're going to use in this study in a paragraph or two? It would be great to provide more information regarding this relatively poorly known area of the world in terms of tectonic settings and volcanic activity. I would also perhaps even a create a separate section for the volcano history.

Going through the very interesting table 1, I noticed that eruptions are particularly short (a few days). You often refer to Kelud to interpret your results and understand the hazards at Awu which seem totally relevant to me. In our recent paper (Caudron et al., 2015, GRL), we noticed that Kelud had very short but intense eruption and hence reasonable VEI ( $\sim$ 4). Do you think this is the case at Awu? Any way to compute the intensity along with the VEI which may better reflect explosivity?

As clearly stated in the paper, another manuscript is being considered for publication in GRL. It would be interesting to explain how they differ as 1 table and several figures are found in both manuscripts (https://www.essoar.org/doi/pdf/10.1002/essoar.10501997.1).

L.167-169: I'm a bit lost here. You basically explain that 27 MW of radiant flux would be sufficient to evaporate all the incoming water (without infiltration) in maximum 8 hrs. This is convincing but why is this coherent with the drying out prior to the 1992 eruption? We don't know how fast it did evaporate since there is no date mentioned in Table 1, and the volume was more than 18 times larger than the one you mention on I.155. Similarly how does that support the drying out prior to the 2004 eruption? You may expect more heat to be transferred to the system prior to eruption but I'm a bit lost concerning the take-home message here. The section title Transition of heat to the surface controls the water accumulation is confusing to me at this stage. The fact that
you did not observe any water in 2015 could simply be explained by the evaporation am I right? So the water accumulation is not simply controlled by heat coming from below.

L.179: I don't understand what supports the statement regarding lava domes emplacement without explosive magma-water interactions?

L.186-187: this is wrong. The 2014 Kelud eruption occurred after 7 years following the dome emplacement. Question: my understanding was the dome quickly grew at Kelud, within a few months or so, then completely stop growing? Is it the case for Awu?

L.196: other mechanisms exist. Just to keep the parallel with Cassidy et al. 2019 (G3) Kelud, suaaest internal trigaering: https://agupubs.onlinelibrary.wiley.com/doi/full/10.1029/2018GC008161. Another may relate to permeability reduction due to alteration at dome-forming volcanoes (Heap et al., 2019). I feel you should discuss these options in detail taking into account their knowledge of the Awu system.

L.215-220: this message is an important one but need to be supported better. You seem to imply that the explosivity of the past vigorous eruptions is related to magma-water interactions. Am I right? The example of Kelud 2007 vs 2014 shows that the water had only a negligible effect on the explosivity. Could you comment/elaborate on this?

Minor questions L.24: what is a little know volcano?

L.28: what are global impacts? L.40: It would be interesting for the reader to explain why/how some injections in the stratosphere lead to a cooling while other produce a warming. Just in 1-2 sentences

L.75: was the Multi-GAS deployed on the dome? The arrow on figure 2. You mention different locations in the text but there is only 1 arrow in the figure

L.101: this low frequency is interesting. What would create a 0.3 Hz pulsation? Are
there other peaks at other frequencies?

L.159: what is the ambient temperature considered?

L.187-188 : which volcano are you referring to?

Technical Corrections L.26: reference for the extension to the sea bed is missing L.44: casualties L.97: it should be Figure 4 L.129: Cashmana? L.133: order of references? L.136: It is also Section 4.3. L.168: will no longer be sufficient L.176: the Kelud crater lake was not huge (2 million m3). L.176-177: But it is I.193: destabilize L.195: megapascals L.206: suggestion: rephrase this sentence. 'arc. 18 eruptions occurred over the last 3.5 centuries, including...' L. 208: Earth

Table 1: 1892: Why do you capitalize Tsunami and Pyroclastic here? And you don't use bold style for the number of victims. Make sure to be consistent throughout the table Figure 1: Great figure. There is a A, but no B or C. A color scale is missing for the 3D map on the right side. The bold labels on the map are a bit hard to read. Figure 6: can't find the GVP, 2013. I'd would also use consistent label sizes and perhaps change the white color to black for the 28/07/2015 photo. Figure 7: typo: circulations

---

## Referee Comment (RC2) · Caroline Bouvet de Maisonneuve (Referee) · 13 Apr 2020

In this paper, the authors present a broad overview of Mount Awu in Indonesia; it's eruptive history, magma composition, current degassing mechanism and potential hazard. It is a combination of interesting thoughts and observations but there is a lack of focus and thus it reads more like an almost random collection of ideas rather than a targeted study. It is a largely unstudied, yet hazardous volcano, thus this attempt at characterizing it is valid and important. Below are some suggestions and key points to be addressed. Specific comments regarding the text were inserted in the pdf directly.

1. In the introduction, please provide more information about the purpose of this study

and the focal point of the manuscript. Did you compile all the info in Table 1? If so, it would be worth highlighting explicitly. Why did you obtain whole-rock analyses? Was it just to know the average composition of Awu lavas (assuming that the current dome is representative), or was it needed to compute gas ratios? Why did you analyse the volatile flux and gas ratios, i.e. how does it fit with the rest of the data presented here and why report it here rather than in Bani et al., submitted (what is the title and where was it submitted?)? You have to tie in these types of information a bit better to strengthen this contribution.

2. The interpretation of the geochemical data is overstretched. From your 2 whole-rock analyses, you cannot conclude that the peculiar tectonic setting of Sangihe is at the origin of the recurring strong activities at Awu. There are recurring violent eruptions at other volcanoes in Indonesia or the rest of the world, which are in very different tectonic settings, and Kelud (cited in this paper as an analogue of Awu's alternating dome – explosive activity) is a good example. Please revise the interpretation, and provide more information regarding the sampling location, sample descriptions, and analytical methods.

3. The flow of the text is good but the English can be improved. I made some suggestions in the PDF.

Please also note the supplement to this comment:
https://www.nat-hazards-earth-syst-sci-discuss.net/nhess-2020-27/nhess-2020-27-RC2-supplement.pdf

**Supplement:**

[Figure]

**Insights into the recurrent energetic eruptions on Awu, one of the deadliest volcano 
[revised manuscript text omitted]

115   high $CO_2$/$^3$He ratios (64-180 x10$^9$) as well as high $\delta^{13}$C ($\geq$-2‰) suggesting slab contribution into the magmatic fluids at Awu. Clor et al. (2005) further point out anomalous high $N_2$/He (2852) coupled with low $\delta^{15}$N (3.3%) suggesting increased slab contribution, possibly by slab melting as collision stalls the progress of the subducting plate and allows it to become superheated (Peacock et al., 1994). This is supported by the slow-down of the subduction rate as evidenced by seismic studies (McCaffrey, 1983; Pubellier et al., 1991; Zhang et al., 2017). This particular double subduction and arc-

120   arc collision have rendered the slab prone to melting that subsequently produces the magmatic source behind the recurrent strong eruptive activities on Awu. The mechanism also contributes to unusual slab carbon delivery into the mantle as highlighted by the extremely elevated $CO_2$ (Bani et al. submitted).

**4.2 A conduit plug**

125   Lava domes are formed when viscous lava extruded to the surface effusively then pile up around the vent. Such phenomena involve complex processes, including crystallization, bubble nucleation, growth, coalescence and out-gassing, bulk magma deformation, crack propagation and healing (e.g., Ashell et al., 2015 and ref therein). It is the competition between these processes that either promote or prevent degassing, leading to explosions or stability of a lava dome (Klug and Cashmana, 1996; Takeuchi et al., 2005; Mueller et al., 2008). On Awu, the $SiO_2$ content of the lava

130   dome higher than 50 wt% as well as the perfect semi-spherical morphology of ~1.3x10$^7$ m$^3$ that extended from the middle of the crater suggests an endogenous growth that generally inflates the dome carapace through magma injection at depth. In such case, lava domes are known to induce variable porous and brecciated carapace surrounding a denser and coherent interior (Fink et al., 1992; Wadged et al., 2009, Ashell et al., 2015; Newhall and Melson, 1983) suitable to form a plug in the upper conduit (Watts et al. 2002). The radiant thermal energy around the lava dome is much higher

135   than the heat release from the dome representing 79% of the total 27 MW from the crater. Only ~6 MW is released through the lava dome itself. It also around the dome that much of the gas is released to the atmosphere (Figure 5). The hottest surfaces also correspond to the main degassing points which suggest that heat is rather sustained by fluid circulations around the dome. With a conduit plug, the gas released at depth is thus forced to the periphery of the lava dome (Fig.5), similar to other dome-forming systems, including Rokatenda (Primulyana et al., 2018), Lascar (Matthews

140   et al., 1997) or Soufriere Hills (Sparks, 2003).
It is thus obvious that the existence of a conduit plug may constitute a barrier to the gas flow, suitable for rapid pressure built up with new magma injection, a situation that can strongly contribute to the vigorous explosions on Awu.

[Figure]

**4.3 Transition of heat to the surface controls the water accumulation**

145    Out of the 17 recorded eruptive activities on Awu, 11 were tagged as phreatic and 6 other eruptions were phreatomagmatic (Table 1). It is thus unambiguous that water played a major role in Awu volcanic activity. Indeed, with an average annual rainfall of 3500 mm (Stone, 2010) and a crater area of 1.5 km$^2$, the Awu summit is likely to accommodate $5.2 \times 10^6$ m$^3$ of water each year. Given that there is no visible water outlet from the crater, one can expect

150    water accumulation and strong infiltration into the hydrothermal system which may then subsequently contribute to phreatic eruptions. But as highlighted in figure 6, surface water was not always present in Awu's crater. Crater lake existed in 1922, 1973 and 1995 whilst in 1931 and 1979 crater lake co-existed with a lava dome. In July 2015 (this fieldwork) there was no water in the crater and a lava dome occupied the central part of the crater. July is among the driest month of the year, however, the average monthly rainfall on Sangihe Island doesn't fall below 130 mm (Stone,

155    2010). Hence one can expect a cumulative water volume of at least $195 \times 10^3$ m$^3$ (equivalent to $7 \times 10^6$ moles or $1.3 \times 10^8$ g, using PV=nRT) into Awu's crater on that period of the year. But the absence of water as observed in July indicate that the water was efficiently infiltrated and evaporated away. In theory, if we assume that the infiltration is negligible, then it requires a heat energy of $8.0 \times 10^{11}$ joules (using $mC_p\Delta T$; m is the water mass, Cp is the water's specific heat capacity, $\Delta T$ is the difference between boiling and ambient temperature) to bring the above volume to the evaporation temperature

160    (100°C) and another $4.6 \times 10^9$ joules (using mL; L is the latent heat of vaporization) to convert it into water vapor. A total $8.1 \times 10^{11}$ joules is thus sufficient to dry out the July incoming water volume. With 27 MW of radiant flux from the crater, equivalent to $2.7 \times 10^7$ J s$^{-1}$, only 8 hours is necessary to heat the $7 \times 10^6$ moles of water from 16 °C to the evaporation temperature and transform it to water vapor. This duration should be considered maximum as the portion of water infiltration is ignored. Nevertheless, the above simple calculation suggests that the heat transfer to the surface from the

165    magmatic source is largely sufficient to evaporate out the water and thus controlling the water accumulation in Awu's crater. This is coherent with the 95% water loss by evaporation from the $3.5 \times 10^6$ m$^3$ of the lake volume before to the 1992 eruption (Table 1). Similarly, in 2004 the lake water progressive dried out before the eruption. But if that the cooling trend in the crater continues, then ultimately the heat will no longer sufficient to dry out the incoming water from the rainfall. Water may then accumulate to form a crater lake as already observed in the past.

170

**4.4 The Hazardous situation**

         The contact of liquid water with a hot surface is widely accepted as a process that can trigger explosive water-magma interactions (Wohletz, 1986; 2002; Zimanowski et al., 1995; Thiéry and Mercury, 2009). However, according to a review of historical eruptions through volcanic lakes, ~2% involved relatively passive growth of subaqueous to

175    emergent lava domes (Manville, 2015). This was the case on Kelud volcano (Java) in 2007 where a lava dome emerged in the middle of a huge crater lake without any vigorous explosion as suspected (Hidiyati et al., 2009). But only 7 years later, in 2014 that a VEI 4 eruption was witnessed on the volcano (Kristiansen et al., 2014). On Awu given the limited information on its past activities, it is difficult to provide a rigorous description of the successive lava dome emplacement. However, the coexistence of lava dome and crater lake in 1931 and 1979 were preceded by crater lake

180    solely and no eruption in between (Fig.6). Such sequence suggests that Awu lava domes possibly occurred through the crater lake without explosive water-magma interactions. Further, the cooling and the relatively small and steady volume

of Awu's lava dome on a flat crater floor represent less probability for a dome collapse, contrasting with other domes were the average rate of dome growth is approximately $10^4$ m$^3$ day$^{-1}$ with a mean volume of $5\times10^7$ m$^3$ and some with unstable slopes (Newhall and Melson, 1983). The hazardous situation on Awu is thus more related to the presence of the

185 conduit plug and a crater lake since this latter may further increase the potential of violent eruptions (Sheridan and Wohletz 1983; Wohletz 1986). Similar to Kelud 2014 eruption, the vigorous VEI 4 eruption witnessed on Awu in 1966 occurred 35 years following the lava dome emplacement. Also, the 1992 eruption has pulverized a lava dome formed 13 years earlier. The current lava dome on Awu has developed 16 years ago, just after the 2004 eruption. The prevalence of $H_2S$ over $SO_2$, the low $SO_2$ emission budget of 13 t/d and the low equilibrium temperature of circa 380 °C obtained by

190 resolving together the $SO_2/H_2S$ vs. $H_2/H_2O$ redox equilibria (see methodology in Aiuppa et al., 2011; Moussallam et al., 2017) indicate that the current activity on Awu is sustained by a degassed magma. Hence, assuming that the processes that lead to vigorous eruption will repeat themselves then the subsurface volcanic system has to develop more than 3.1 MPa of pressure (Pressure (Pa) = F/Area (m$^2$); F (N) = masse (kg) x 9.8 (m/s$^2$); density of 2700 kg/m$^3$) to destabilized the $3.5\times10^{10}$ kg of lava dome. The total pressure than can be developed in the conduit or reservoir generally reaches tens of

195 magapascals (e.g., Gudmundsson, 2012), largely sufficient to clear-up the conduit and pulverize the lava dome. Thus a new magma injection into the reservoir may be necessary to induce a rapid pressure buildup, capable to pulverize the lava dome and the conduit plug. This latter scenario may constitute an opening phase of volcanic eruption, creating a favorable condition for a rapid external lake water injection that could lead to magma-water interaction. Events of particularly high intensity described as phreato-Plinian eruptions can only be explained with the involvement of surface

200 water in the eruption dynamics (Areva et al., 2018). A presence of water in Awu's crater may thus increase the potential of intense eruptive manifestations, a possible scenario behind the past vigorous eruptions. The most hazardous situation on the Awu volcano would be the new magma injection into the reservoir beneath a conduit plug and a sufficient water volume in the crater lake (Fig.7).

205 **5 Conclusion**

Awu is the northernmost active volcano of the Sangihe arc with 18 eruptions over the last 3.5 centuries, including 2 with VEI 4 and 3 with VEI 3 with local, regional and even worldwide impacts. Awu is also one of the deadliest volcano on earth with a cumulative dead toll of 11048 following its recurrent eruptive activity. 
[revised manuscript text omitted]

---

## Author Comment (AC1) · 8 May 2020

We would like to thank Dr. Corentin Caudron (RC1) for the detailed review of this manuscript. Here we provide our response to the remarks, comments, and suggestions.

Specific comments (SC):

SC 1: The introduction just documents past eruptions. It would be interesting to briefly introduce your aim and which methods you're going to use in this study in a paragraph or two? It would be great to provide more information regarding this relatively poorly

known area of the world in terms of tectonic settings and volcanic activity. I would also perhaps even a create a separate section for the volcano history.

Response We agree that the introduction part of this manuscript is too long and the objective of the work doesn't stand out sufficiently. The new version of the introduction is divided into sub-sections which should allow readers to have general ideas about this Awu volcano. The objective of this manuscript is to highlight the intense eruptive character of this volcano and provide insights into the possible mechanisms that fueled the deadly energetic eruptions. The geodynamic context is summarized as suggested and figure 1 is modified to better illustrate the peculiar geodynamic context dominated by the unique present-day example of arc-to-arc collision.

Figure 1 full caption. Awu volcano is the northernmost active volcano of the Sangihe arc (A). It occupies the northern portion of Sangihe island (B). 3D map from https://maps-for-free.com. Sangihe and Halmahera arcs constitute the present-day example of arc-to-arc collision (C). The Molucca Sea Plate that existed between the two arcs is now sinking deeper beneath the Molucca Sea. Awu's crater is currently occupied by a lava dome (D). Note the person circled in red for scale.

SC 2: Going through the very interesting table 1, I noticed that eruptions are particularly short (a few days). You often refer to Kelud to interpret your results and understand the hazards at Awu which seem totally relevant to me. In our recent paper (Caudron et al., 2015, GRL), we noticed that Kelud had very short but intense eruption and hence reasonable VEI (4). Do you think this is the case at Awu? Any way to compute the intensity along with the VEI which may better reflect explosivity?

Response Awu is poorly studied and this manuscript is among the rarely available if any. Thus more work is needed to gain more information, particularly the mass discharges, the ash dispersal, plume heights, etc. Such information may help to compute the past eruptions intensities to better characterize the strength and the hazards behind each event. This is beyond the scope of this manuscript. However we agree on
the short duration events and have added Caudron et al. (2015) for reference.

SC 3: As clearly stated in the paper, another manuscript is being considered for publication in GRL. It would be interesting to explain how they differ as 1 table and several figures are found in both manuscripts (https://www.essoar.org/doi/pdf/10.1002/essoar.10501997.1).

Response Yes there is another manuscript on Awu submitted to GRL in which the location figure, the crater figure, and the table of eruptive history are the same. However, in contrast to the manuscript considered in NHESS, the second manuscript focuses on the gas emission on Awu and more specifically on the abnormal CO2 emission. The manuscript highlighted this CO2 rich gas and provide some hypothesis on its source whilst this NHESS manuscript focuses on the volcano and its intense eruptive activity. We prefer to develop fully these two topics in separate manuscripts.

SC 4: L.167-169: I'm a bit lost here. You basically explain that 27 MW of radiant flux would be sufficient to evaporate all the incoming water (without infiltration) in maximum 8 hrs. This is convincing but why is this coherent with the drying out prior to the 1992 eruption? We don't know how fast it did evaporate since there is no date mentioned in Table 1, and the volume was more than 18 times larger than the one you mention on I.155. Similarly how does that support the drying out prior to the 2004 eruption? You may expect more heat to be transferred to the system prior to eruption but I'm a bit lost concerning the take-home message here. The section title Transition of heat to the surface controls the water accumulation is confusing to me at this stage. The fact that you did not observe any water in 2015 could simply be explained by the evaporation am I right? So the water accumulation is not simply controlled by heat coming from below.

Response Thank you for this remark. The subtitle is changed to avoid confusion and reflect the content better. The new sub-title is "The heat transfer to the surface controls the water accumulation". With the high annual rainfall of 3500 mm and the 3.5 million
cubic meter of water in the crater, the solar heating, combined with the heat provided by the atmospheric radiation may not be sufficient to evaporate out the 95 % the lake water, if the heat input from a shallow magma is negligible. The main message here is the role of heat provided to the surface. If the increase of heat flux can lead to water lake evaporation, the cooling of the crater surface can in contrast allow water to accumulate. Hence with the current cooling trend in the crater, one would expect that ultimately the heat supply to the surface and the solar heating will no longer be sufficient to dry out the incoming water from the rainfall. Water may then accumulate to form a new crater lake, as already seen in the past.

SC 5: L.179: I don't understand what supports the statement regarding lava domes emplacement without explosive magma-water interactions?

Response By the time the viscous lava reaches the surface, its temperature could be as hot as 600°C. Thus if the water comes into contact with such hot magmatic body one can expect intense magma-water interaction and eventual phreatic eruption. However, alternatively, during ascent, the crystallizing magma body may release much of its gas and the carapace surface temperature can rapidly drop below 100°C once it reached the surface (Sherrod et al., 2008). In such a scenario the dome may passively emerge through a crater lake without explosive magma-water interactions.

SC 6: L.186-187: this is wrong. The 2014 Kelud eruption occurred after 7 years following the dome emplacement. Question: my understanding was the dome quickly grew at Kelud, within a few months or so, then completely stop growing? Is it the case for Awu?

Response Thanks for this remark. We reformulate the text to better express the delay between dome emplacement and violent eruption. Similar to Kelud, the lava dome on Awu has rapidly reached its current size then completely stopped from growing. This is now mentioned in the text.

SC 7: L.196: other mechanisms exist. Just to keep the paral-
lel with Kelud, Cassidy et al. 2019 (G3) suggest internal triggering: https://agupubs.onlinelibrary.wiley.com/doi/full/10.1029/2018GC008161. Another may relate to permeability reduction due to alteration at dome-forming volcanoes (Heap et al., 2019). I feel you should discuss these options in detail taking into account their knowledge of the Awu system.

Response We agree with this remark and many thanks for the references. Other mechanisms that are likely to trigger an explosion are now included in the manuscript as suggested, including the second crystal nucleation and rapid crystallization of a degassed magma, as well as the reduction of lava dome permeability with the hydrothermal processes.

SC 8: L.215-220: this message is an important one but need to be supported better. You seem to imply that the explosivity of the past vigorous eruptions is related to magma water interactions. Am I right? The example of Kelud 2007 vs 2014 shows that the water had only a negligible effect on the explosivity. Could you comment/elaborate on this?

Response The similarity that we highlight between Awu and Kelud focuses on the passive emplacement of the lave dome through crater lake and the time delay between the dome emplacement and the VEI 4 eruptions. In contrast we consider that there was a coexistence of crater lake and lava dome when the VEI 4 eruption occurred on Awu which is not the case on Kelud. The triggering mechanism of kelut 2014 eruption was the second crystal nucleation event and the subsequent rapid crystallization at shallow depth that led to over-saturation of the source with intense diffusion of volatiles and growth of bubbles. Investigate the triggering mechanism is beyond the scope of our work. We simply quote the common process – the injection of a new magma – as the triggering event. Thanks for the remark, we now include in the manuscript other mechanisms that can trigger the eruptive activity on Awu, including the second crystal nucleation and the acidic-sulphate alteration processes.
**Minor questions (MC)**

MC 1: L.24: what is a little know volcano? It should be little known - now corrected

MC 2: L.28: what are global impacts? L.40: It would be interesting for the reader to explain why/how some injections in the stratosphere lead to a cooling while other produce a warming. Just in 1-2 sentences. In general massive sulfate aerosol injections into the stratosphere, increase the stratospheric aerosol's optical depth leading to a reduce of surface temperature. However, major tropical eruptions can produce asymmetric stratospheric heating that can ultimately enhance warming on some regions and cooling on others. We add reference to Robock (2015) for further detail on asymmetric stratospheric heating.

MC 3: L.75: was the Multi-GAS deployed on the dome? The arrow on figure 2. You mention different locations in the text but there is only 1 arrow in the figure. Thanks for this remark – indeed we deployed the multi-GAS on 3 points but only one of them is less diluted and considered as representative of the system. It is now clearly mentioned.

MC 4: L.101: this low frequency is interesting. What would create a 0.3 Hz pulsation? Are there other peaks at other frequencies? This figure is provided to highlight the dynamic of the degassing. The mechanism in the hydrothermal system that lead to this dynamic is not developed here as longer time series may be required in order to investigate the process.

MC 5: L.159: what is the ambient temperature considered? There was no available meteorological data for Awu summit, thus 16°C obtained with the IR is used as ambient value. It is now indicated in the text.

MC 6: L.187-188 : which volcano are you referring to? The sentence is modified and refers to Awu 1992 eruption.

MC 7:Technical Corrections L.26: reference for the extension to the sea bed is missing thanks – a link is added for reference (www.opendem.info). MC 8: Table 1: 1892: Why
do you capitalize Tsunami and Pyroclastic here? And you don't use bold style for the number of victims. Make sure to be consistent throughout the table – characters are now homogenized.

MC 9: Figure 1: Great figure. There is a A, but no B or C. A color scale is missing for the3D map on the right side. The bold labels on the map are a bit hard to read. Now corrected in new Figure 1.

MC 10: Figure 6: can't find the GVP, 2013. I'd would also use consistent label sizes and perhaps change the white color to black for the 28/07/2015 photo. - the GVP 2013 reference is now added and the figure labels are homogenized.

Typo: L.44: casualties – corrected L.97: it should be Figure 4 - corrected L.129: Cashmana? - corrected L.133: order of references? - corrected L.136: It is also - corrected L.168: will no longer be sufficient - corrected L.176: the Kelud crater lake was not huge (2 million m3). - corrected L.176-177: But it is - corrected I.193: destabilize - corrected L.195: megapascals - corrected L.206: suggestion: rephrase this sentence. 'arc. 18 eruptions occurred over the last 3.5 centuries, including: : :' - the sentence is modified L. 208: Earth - corrected Figure 7: typo: circulations - corrected

---

## Author Comment (AC3) · 11 May 2020

Please refer to the separate AC1 to access the responses to RC1. Posted by Philipson Bani on 08 May 2020.
* * *

---

## Author Response (AR1)

Editor Decision: Reconsider after major revisions (further review by editor and referees) (09 Jun 2020) by Giovanni Macedonio

**Comments to the Author:**

I wish to thank the reviewers for their comments and suggestions. According to the comments of the reviewers and the authors' answers, I invite the authors to submit a revised version of the manuscript. Moreover, I have a concern regarding the publication of the material already submitted for publication on another journal (Table 1 and Figure 1). It is quite unlikely that published figures can be best re-used for publication. In this case there should be a proper citation along with the figure, but if it is not necessary the respective figure/table should be left out and referred to as a citation. Else, the figures should be changed so that they fit the needs of the manuscript, to avoid possible self-plagiarism.

**Resp.**

A new version of the manuscript is now provided, taking into account all the corrections, suggestions and comments from the two reviewers. Please be informed that the separate paper reporting on the anomalous $CO_2$ content in Awu's gas was declined for publication in GRL. The topic and study are of interest, but there remains important question as to the validity of the conclusions, especially given the short time window where gases were measured and the lack of evidence for that window being representative of the long-term behavior of the system. The paper will be re-submitted as additional data highlight acquired 15 years earlier highlight the some conclusion. **The new version of the CO2 manuscript will no longer include the figure 1 and the Table 1. It is more appropriate that they appear in this NHESS manuscript.**

**Reviewer 1: Corentin Caudron**

**Q1**

The introduction just documents past eruptions. It would be interesting to briefly introduce your aim and which methods you're going to use in this study in a paragraph or two? It would be great to provide more information regarding this relatively poorly known area of the world in terms of tectonic settings and volcanic activity. I would also perhaps even a create a separate section for the volcano history.

**Resp**

Tow sub-sections are introduced in the introduction as suggested, including the "**1.1 Geological setting**" and the "**1.2 Historical activities**".

**Q2**

Going through the very interesting table 1, I noticed that eruptions are particularly short (a few days). You often refer to Kelud to interpret your results and understand the hazards at Awu which seem totally relevant to me. In our recent paper (Caudron et al., 2015, GRL), we noticed that Kelud had very short but intense eruption and hence reasonable VEI (4). Do you think this is the case at Awu? Any way to compute the intensity along with the VEI which may better reflect explosivity?

**Resp**

There as lack on information, particularly the mass discharges and plume heights, thus computing intensity would be difficult without speculating. We however agree on the short duration events and have added Caudron et al. (2015) for reference.

**Q3**

As clearly stated in the paper, another manuscript is being considered for publication in GRL. It would be interesting to explain how they differ as 1 table and several figures are found in both manuscripts (https://www.essoar.org/doi/pdf/10.1002/essoar.10501997.1).

**Resp**

Yes there is another manuscript on Awu submitted to GRL in which the location figure, the crater figure and the table of eruptive history is the same. But in contrast the manuscript considered in NHESS, the GRL manuscript focuses on the gas emission on Awu and more specifically on the abnormal $CO_2$ emission.

**Q4**

L.167-169: I'm a bit lost here. You basically explain that 27 MW of radiant flux would be sufficient to evaporate all the incoming water (without infiltration) in maximum 8 hrs. This is convincing but why is this coherent with the drying out prior to the 1992 eruption? We don't know how fast it did evaporate since there is no date mentioned in Table 1, and the volume was more than 18 times larger than the one you mention on l.155. Similarly how does that support the drying out prior to the 2004 eruption? You may expect more heat to be transferred to the system prior to eruption but I'm a bit lost concerning the take-home message here. The section title Transition of heat to the surface controls the water accumulation is confusing to me at this stage. The fact that you did not observe any water in 2015 could simply be explained by the evaporation am I right? So the water accumulation is not simply controlled by heat coming from below.

**Resp**

Thank you for this remark. We change the section title to avoid confusion. The new sub-title is **"The heat transfer to the surface controls the water accumulation".** However, given the high annual rainfail of 3500 mm, and the 3.5 million cubic meter of water in the crater, the solar heating, combined with the heat provided by the atmospheric radiation may not be sufficient to evaporate out the 95 % the lake water. Thus the heat input from a shallow magma may contribute to the evaporation.

If the increase of heat flux can lead to water lake evaporation, the cooling of the crater surface can in contrast will allow water to accumulate. Hence with the current cooling trend in the crater, one would expect that ultimately the heat supply to the surface will no longer sufficient to dry out the incoming water from the rainfall. Water may then accumulate to form a new crater lake, as already seen in the past.

**Q5**

L.179: I don't understand what supports the statement regarding lava domes emplacement without explosive magma-water interactions?

**Resp**

By the time it reaches the surface the viscous lava could be as hot as 600°C. Thus if water come into contact with such hot magmatic body one can expect explosive magma-water interaction. However, during ascent, the crystallizing magma may release much of its gas and the carapace surface temperature can rapidly cooled below 100°C once reached the surface (Sherrod et al., 2008). In such scenario the dome may passively emerged through a crater lake without explosive magma-water interactions.

**Q6**

L.186-187: this is wrong. The 2014 Kelud eruption occurred after 7 years following the dome emplacement. Question: my understanding was the dome quickly grew at Kelud, within a few months or so, then completely stop growing? Is it the case for Awu?

**Resp**

Thanks for this remarks. We now reformulate the text to better express delay between dome emplacement and violent eruption. Yes the dome has rapidly reached its current size then completely stop growing. This is now mentioned in the text.

**Q7**

L.196: other mechanisms exist. Just to keep the parallel with Kelud, Cassidy et al. 2019 (G3) suggest internal triggering: https://agupubs.onlinelibrary.wiley.com/doi/full/10.1029/2018GC008161. Another may relate to permeability reduction due to alteration at dome-forming volcanoes (Heap et al., 2019). I feel you should discuss these options in detail taking into account their knowledge of the Awu system.

**Resp**

Other mechanisms that can trigger explosions are now included into the manuscript as suggested, including the second crystal nucleation and rapid crystallization of a degassed magmas, as well as the reduction of lava dome permeability with the hydrothermal processes.

**Q8**

L.215-220: this message is an important one but need to be supported better. You seem to imply that the explosivity of the past vigorous eruptions is related to magma water interactions. Am I right? The example of Kelud 2007 vs 2014 shows that the water had only a negligible effect on the explosivity. Could you comment/elaborate on this?

**Resp**

The similarity that we highlight between Awu and Kelud focuses on the passive emplacement of the lave dome through crater lake and the time delay between the dome emplacement and the VEI 4 eruptions. In contrast we consider that there was a coexistence of crater lake and lava dome when the VEI 4 eruption occurred on Awu which is not the case on Kelud. The triggering mechanism of kelut 2014 eruption was the second crystal nucleation event and it was the subsequent rapid crystallization at shallow depth that led to over-saturation of the source with intense diffusion of volatiles and growth of bubbles. That part is beyond the scope of our work and thus we simply quote the common process – the injection of a new magma – as the triggering event. Thanks for the remark, we now include in the manuscript other mechanisms that can trigger the eruptive activity on Awu, including the second crystal nucleation and the acidic-sulphate alteration processes.

Minor questions
L.24: what is a little know volcano?
It should be little known - thanks

L.28: what are global impacts? L.40: It would be interesting for the reader to explain
why/how some injections in the stratosphere lead to a cooling while other produce a
warming. Just in 1-2 sentences
In general massive sulfate aerosols injection into the stratosphere, increases the stratospheric aerosol's optical depth leading to a reduce of surface temperature. However, major tropical eruptions can produce asymmetric stratospheric heating (Robock, 2015) that can ultimately enhance warming on some regions and cooling on others.

L.75: was the Multi-GAS deployed on the dome? The arrow on figure 2. You mention different locations in the text but there is only 1 arrow in the figure.
Thanks for this remark – indeed we deployed the multi-GAS on 3 points but only one of them is consider less diluted that we consider more representative of the system and presented in this manuscripts. The manuscript is now adjusted.

L.101: this low frequency is interesting. What would create a 0.3 Hz pulsation? Are there other peaks at other frequencies?
This figure is provided to highlight the dynamic of the degassing. The mechanism behind is not developed here as it is beyond the scope of this work.

L.159: what is the ambient temperature considered?
There was no available meteorological data for Awu summit, thus 16°C obtained with the IR is used as ambient value. It is now indicated in the text.

L.187-188 : which volcano are you referring to?
The sentence refers to Awu 1992 eruption and is now better referenced.

Technical Corrections L.26: reference for the extension to the sea bed is missing
thanks – a link is added for reference (www.opendem.info)

L.44: casualties – corrected

L.97: it should be Figure 4  - corrected

L.129: Cashmana?  - corrected

L.133: order of references?  - corrected

L.136: It is also - corrected

L.168: will no longer be sufficient - corrected

L.176: the Kelud crater lake was not huge (2 million m3).  - corrected

L.176-177: But it is - corrected

l.193: destabilize  - corrected

L.195: megapascals  - corrected

L.206: suggestion: rephrase this sentence. 'arc. 18 eruptions occurred over the last 3.5 centuries, including: : :'
- corrected

L. 208: Earth - corrected

Table 1: 1892: Why do you capitalize Tsunami and Pyroclastic here? And you don't use bold style for the number of victims. Make sure to be consistent throughout the table - corrected

Figure 1: Great figure. There is a A, but no B or C. A color scale is missing for the3D map on the right side. The bold labels on the map are a bit hard to read. - corrected

Figure 6: can't find the GVP, 2013. I'd would also use consistent label sizes and perhaps change the white color to black for the 28/07/2015 photo.  - corrected

Figure 7: typo: circulations

**Revewer 2: Caroline Bouvet de Maisonneuve**

Main Questions (MQ)

**MQ 1:** In the introduction, please provide more information about the purpose of this study and the focal point of the manuscript.

Response:
The objective of this manuscript is to highlight the intense eruptive character of Awu volcano and provide insights into the possible mechanisms that fueled the deadly energetic eruptions. We thus adjust the title to better reflect the objective of this work. The title is changed to "**Insights into the recurrent energetic eruptions that drive Awu among the deadliest volcanoes on earth**".

Did you compile all the info in Table 1? If so, it would be worth highlighting explicitly.

Response:
Yes we did and now mention it in the text.

**MQ 2:** Why did you obtain whole-rock analyses? Was it just to know the average composition of Awu lavas (assuming that the current dome is representative), or was it needed to compute gas ratios?

Response:
The bulk rock analyses are intend to provide an idea about the lava dome composition and also to provide readers with as much information as possible about this little know volcano.

**MQ 3:** Why did you analyse the volatile flux and gas ratios, i.e. how does it fit with the rest of the data presented here and why report it here rather than in Bani et al., submitted (what is the title and where was it submitted?)? You have to tie in these types of information a bit better to strengthen this contribution.

Response:
Gas composition and emission rates provide important information about the magma source behind the observed activity. As mentioned in the text, the prevalence of $H_2S$ of $SO_2$ and the low $SO_2$ emission rate indicate a predominant of hydrothermal processes on Awu in the present time. The limited magmatic fluids are thus likely sustained by a degassed magma source, in accord with the low equilibrium temperature of circa 380°C. The above information indicate a continuous cooling tendency in Awu's crater, since 2004.
As for the other manuscript (*Bani et al. submitted*), it was submitted to GRL and focuses more specifically on the $CO_2$-rich gas from Awu and the possible source mechanisms. In contrast, this NHESS manuscript focuses on the Awu volcano and its

intense eruptive activities. We prefer to develop fully these two topics in separate manuscripts. We now provide a full reference to the manuscript submitted to GRL.

**MQ 4:** The interpretation of the geochemical data is overstretched. From your 2 whole-rock analyses, you cannot conclude that the peculiar tectonic setting of Sangihe is at the origin of the recurring strong activities at Awu. There are recurring violent eruptions at other volcanoes in Indonesia or the rest of the world, which are in very different tectonic settings, and Kelud (cited in this paper as an analogue of Awu's alternating dome – explosive activity) is a good example. Please revise the interpretation, and provide more information regarding the sampling location, sample descriptions, and analytical methods.

Response:
We collected only one fresh (less altered) sample directly on the lava dome, but it was analyzed in two separate laboratories, including Laboratoire Magmas et Volcans (Clermont-Ferrand) and Pôle de Spectrométrie Océan (Brest). We now mention it in the text.
We agree that it is not reasonable to trace the magma source from one sample. However, here our result provides for the first time the composition for the current lava dome on Awu. Data from Morrice et al. (1983) and Hanyu et al. (2012), included in Table 2, are obtained from samples collected in 1978-80 and 1998. The locations of the samples are provided in Hanyu et al. (2012). The current lava dome was formed in 2004. It rapidly reached its current size then completely stopped from growing. We believe it was from the same lava body thus our result may be representative of the lava dome composition.
The triggering mechanism of the 2014 eruption of kelud was the second crystal nucleation event (Cassidy et al., 2019). The subsequent rapid crystallization that followed, has led to over-saturation of the source melt with intense diffusion of volatiles and growth of bubbles. Unfortunately investigating the triggering mechanism is beyond the scope of our work. Thus we simply quote the common process – the injection of a new magma – as the triggering event. We now include in the manuscript other mechanisms that can trigger the eruptive activity on Awu, including the second crystal nucleation and the acidic-sulfate alteration processes.
We agree that the following sentence is not justified in this manuscript:
*"This particular double subduction and arc-arc collision have rendered the slab prone to melting that subsequently produces the magmatic source behind the recurrent strong eruptive activities on Awu. The mechanism also contributes to unusual slab carbon delivery into the mantle as highlighted by the extremely elevated $CO_2$ (Bani et al. submitted)."*
However we still believe the geodynamic context has its role in Awu activity. The above sentence is now replaced by the following sentence: *"This particular double subduction and arc-arc collision have rendered the slab prone to melting (Clor et al., 2005) that subsequently supply the magmatic source beneath Awu volcano.*

**Supplementary edit**
A throughout English edit of the text was kindly provided on a separate file (nhess-202027-RC2-supplement) by this reviewer.
All the correction are integrated in the corrected version of the manuscript.

[revised manuscript text omitted]